# Strong coupling and induced transparency at room temperature with single quantum dots and gap plasmons

Haixu Leng[1], Brian Szychowski[2], Marie-Christine Daniel[2] & Matthew Pelton [1]

Coherent coupling between plasmons and transition dipole moments in emitters can lead to two distinct spectral effects: vacuum Rabi splitting at strong coupling strengths, and induced transparency (also known as Fano interference) at intermediate coupling strengths. Achieving either strong or intermediate coupling between a single emitter and a localized plasmon resonance has the potential to enable single-photon nonlinearities and other extreme light–matter interactions, at room temperature and on the nanometer scale. Both effects produce two peaks in the spectrum of scattering from the plasmon resonance, and can thus be confused if scattering measurements alone are performed. Here we report measurements of scattering and photoluminescence from individual coupled plasmon–emitter systems that consist of a single colloidal quantum dot in the gap between a gold nanoparticle and a silver film. The measurements unambiguously demonstrate weak coupling (the Purcell effect), intermediate coupling (Fano interference), and strong coupling (Rabi splitting) at room temperature.

[1] Department of Physics, UMBC (University of Maryland, Baltimore County), Baltimore, MD 21250, USA. [2] Department of Chemistry & Biochemistry, UMBC (University of Maryland, Baltimore County), Baltimore, MD 21250, USA. Correspondence and requests for materials should be addressed to M.P. (email: mpelton@umbc.edu)

Coupling optical emitters to plasmon resonances in metal nanostructures has long been investigated as a means of increasing their spontaneous emission rates[1]. This occurs for weak coupling between the emitter and plasmon; for sufficiently strong coupling, the system is expected to undergo Rabi splitting into new, hybrid modes[2]. These weak- and strong-coupling regimes for plasmon–emitter coupling are equivalent to the weak[3] and strong[4–6] coupling that has previously been observed for single emitters such as epitaxial quantum dots (QDs) coupled to optical cavities. Intermediate coupling strengths, between the weak- and strong-coupling regimes, have been shown to produce a dip in the cavity reflectivity[7]. For both intermediate and strong coupling, the spectrum is highly sensitive to the state of the QD[7,8], which provides a means of manipulating quantum information[9–11]. However, the cavity–emitter coupling strength $g$ scales as $\sqrt{1/V}$, where $V$ is the mode volume of the cavity[12], and the mode volumes of dielectric cavities are restricted to be no less than approximately $(\lambda/2)^3$, where $\lambda$ is the optical wavelength in the material. Since the strong-coupling regime corresponds to $g$ greater than both the cavity and emitter linewidths, and intermediate coupling still requires $g$ comparable to linewidths, achieving these regimes with dielectric cavities requires narrow emitter linewidths, and thus low temperatures.

By contrast, coupling light to plasmonic metal nanoparticles can confine optical energy to volumes well below the diffraction limit[13,14], opening up the possibility of observing intermediate[15–17] and strong[18–20] coupling at room temperature. Earlier reports of room-temperature strong coupling involved ensembles of emitters coupled to plasmons[21–23]. Large coupling strengths were obtained because $g \propto \sqrt{N}$, where $N$ is the number of emitters coherently coupled to a plasmon mode; however, single-photon nonlinearities and quantum gates require strong coupling to individual emitters. Recently, two peaks were observed in the scattering spectra of plasmonic metal nanostructures coupled to single molecules[14] and single QDs[24], and this was taken as evidence of strong coupling. However, irregularities in the structure of the metal nanoparticles can lead to dual scattering peaks, even in the absence of any emitters[25].

Moreover, even without these irregularities, a two-peak structure in the scattering spectrum is not necessarily a sign of Rabi splitting[26–28]. This is illustrated in Fig. 1, where the scattering spectra are calculated according to a coupled-oscillator model[26] for realistic frequencies and linewidths of gap plasmons and room-temperature colloidal QDs, but for different coupling strengths, $g$, between the plasmon and the QD transition. The classical coupled-oscillator model predicts the same spectra as a quantum-mechanical model of a two-level system coupled to a bosonic plasmon field in the linear limit of low excitation intensity, when the QD is well below saturation. Three qualitatively different regimes are seen: for weak coupling, a single scattering peak is observed, nearly unchanged from the scattering from the plasmon resonance in the absence of the quantum dot. For intermediate coupling, still below the threshold for strong coupling, an induced-transparency dip opens up in the scattering spectrum, due to Fano-like interference between the plasmon and QD dipoles. For strong coupling, the scattering peak undergoes Rabi splitting into two separate peaks. In both the intermediate- and strong-coupling regimes, the scattering spectrum has two maxima, making the regimes difficult to distinguish if only scattering is measured[14].

As shown in Fig. 1, however, a measurement of the photoluminescence (PL) spectrum can distinguish between the two regimes. Unlike scattering, PL is an incoherent process, and thus does not display Fano interference. Splitting in the PL spectrum thus occurs only in the strong-coupling regime, and has therefore been recognized as the definitive signature of Rabi splitting[12]. So far, there has been only one report of PL splitting for a single emitter (a QD) coupled to a plasmonic metal nanostructure, but the PL spectrum showed an unexpected four-peak structure[29].

Here, we report straightforward and unambiguous observations of both the strong-coupling regime (Rabi splitting) and the intermediate-coupling regime (Fano interference) for single QDs coupled to individual metal nanostructures, by measuring scattering and PL from the same structures. To achieve ultrasmall mode volumes, we couple QDs to a gap plasmon[14,30]; specifically, we use the plasmon mode localized between a quasi-spherical gold nanoparticle (AuNP) and a silver film.

## Results

**Sample preparation**. The coupled plasmon–QD systems are produced experimentally as illustrated in Fig. 2a. Colloidally synthesized AuNPs and CdSe/CdS QDs are linked covalently through their capping molecules, resulting in a small number of QDs on the surface of each AuNP. Electron-microscope imaging shows that approximately half of the AuNPs have a single QD bound to their surface, and approximately 5% are bound to more than one QD. (See Fig. 2b). The linked particles are deposited on an Ag film by drop-casting; in some cases, a thin silica layer is first deposited on the Ag film.

Although strong coupling has previously been reported in a similar system[14], a finite-element numerical simulation predicts only weak coupling. (See Supplementary Figure 1.) Fitting the calculated spectrum to a coupled-oscillator model gives a coupling strength $g$ of only 10 meV, significantly less than the 100 meV plasmon linewidth. Previous reports also had difficulty using realistic calculations to explain observations of single-emitter strong coupling[14,24]. However, real nanostructures are more complex than the simple geometries used in the calculations. In particular, quasi-spherical metal nanoparticles are always faceted, and these facets can further localize fields, thereby

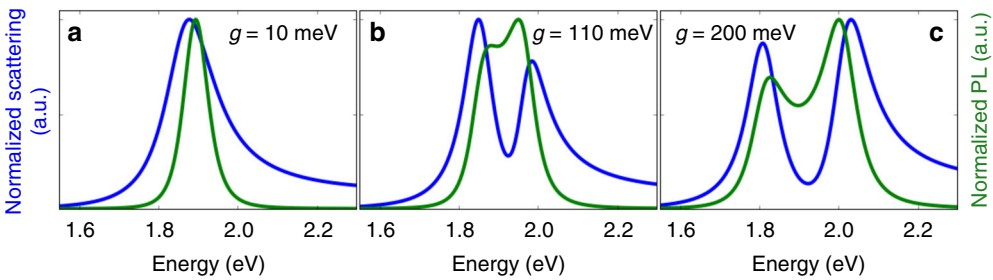

**Fig. 1** Predictions for coupled quantum-dot / gap-plasmon systems in different coupling regimes. **a** Weak coupling, **b** intermediate coupling (Fano interference), **c** strong coupling (Rabi splitting). The predicted scattering spectra (blue) and photoluminescence spectra (green) are calculated according to analytical models using the coupling strengths shown in the insets

increasing coupling strengths. Finite-element simulations predict significantly stronger coupling for a quantum dot located between the silver film and the edge of a facet or the apex of a facet on the gold nanoparticle. This stronger coupling enables the system to enter the intermediate-coupling and strong-coupling regimes. (See Supplementary Figure 1.) The simulations also show that faceting of the nanoparticle leads to strong localization of the fields within the gap in the lateral dimensions, with the electromagnetic energy extending laterally over approximately 10 nm (See Supplementary Figure 2). Since this is less than twice the diameter of the QDs used, at most one QD can be located within the high-field region and coupled to the plasmon resonance.

**Weak-coupling regime**. A single-particle optical microscope is used to measure the scattering and PL, including time-resolved PL, from individual nanostructures. In the majority (approximately 98%) of the cases where PL is observed, both the scattering and PL spectra are single peaks, as illustrated in Fig. 3a. The scattering spectra are comparable to those of AuNPs on an Ag film without any QDs, as shown in Supplementary Figure 3, indicating that the system is in the weak-coupling regime. The PL spectra are comparable to those of isolated QDs (see Supple-

mentary Figure 3), apart from an additional broad background feature, which can be attributed to PL from the metal nanostructure (see Supplementary Figure 4)[31,32]. The narrow linewidth of the peak in the PL spectrum indicates that the PL arises from a single QD: variations in the size and shape between two or more QDs would lead to inhomogeneous broadening of the spectrum.

Fitting the scattering spectrum to a coupled-oscillator model gives $g = 10$ meV, consistent with simulations (see Fig. 3b). The full set of parameters obtained from the fit are given in Supplementary Table 1; using these same parameters to predict the PL spectrum according to Eq. (3) gives a single peak, as shown in Fig. 3b. Although the plasmon–QD coupling leaves the scattering and PL spectra nearly unchanged, it increases the recombination rate in the QD by a factor of 20[1], as shown in Fig. 3c.

**Intermediate-coupling regime**. A small fraction (approximately 1%) of the structures show a clear dip, or induced transparency, in the scattering spectrum; a representative example is shown in Fig. 4a. Fitting the scattering spectrum now gives $g = 100$ meV (see Fig. 4c), corresponding to the intermediate-coupling regime. Using the parameters from the fit to the scattering spectrum leads to a prediction of a single peak in the PL spectrum, located at the minimum of the transparency dip, as expected for near-resonant Fano interference[26]. (See Fig. 4c.) A similar PL spectrum is observed experimentally, with the primary differences being irregularities in the PL background from the metal nanostructure and a high-frequency shoulder that can be attributed to emission from charged-exciton or multiexciton states in the QD[33,34]. We note that the PL linewidth is consistent with the QD linewidth used in fitting the scattering spectrum, and is again consistent with the PL arising from a single QD rather than multiple, inhomogeneously broadened QDs.

This simultaneous measurement of scattering and PL spectra allows Fano interference to be distinguished from geometric properties of the metal nanostructure that could give rise to a double-peak scattering spectrum. It also clearly shows that observing two peaks in the scattering spectrum is not sufficient to demonstrate that a plasmon–emitter system is in the strong-coupling regime.

The interpretation of the observed spectrum as due to single-QD Fano interference is further verified by modifying the QD and plasmon resonance energies. To accomplish this, the system is exposed to intense laser illumination for a fixed amount of time, and scattering and PL measurements are then made under the same low-intensity conditions as the previous measurements. The intense illumination produces an irreversible shift in the PL peak from the QD due to photo-oxidation of the QD surface[35,36], and the plasmon resonance of the metal nanostructure shifts due to small changes in the local structure near the gap[37] (See Supplementary Figure 5). After these spectral shifts, the scattering

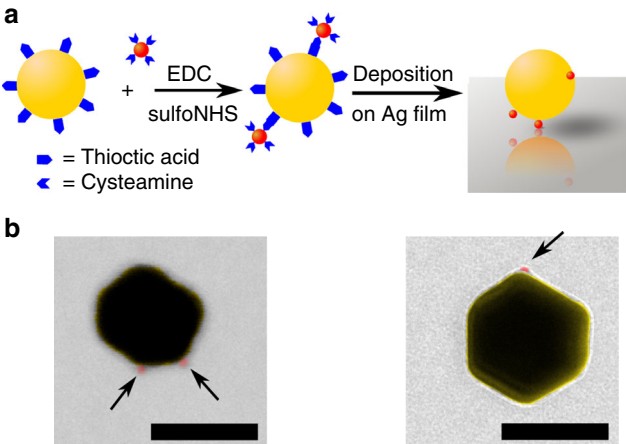

**Fig. 2** Fabrication of coupled quantum-dot / gap-plasmon systems. **a** Illustration of the synthesis process. Quantum dots (red) are linked to gold nanoparticles (yellow) through their capping molecules. The linked assemblies are then deposited on a silver film. **b** Electron-microscope images of linked assemblies. Quantum dots are colored in red and indicated by arrows. The left image was obtained by scanning transmission electron microscopy, and the right image was obtained by transmission electron microscopy. The scale bars are 100 nm

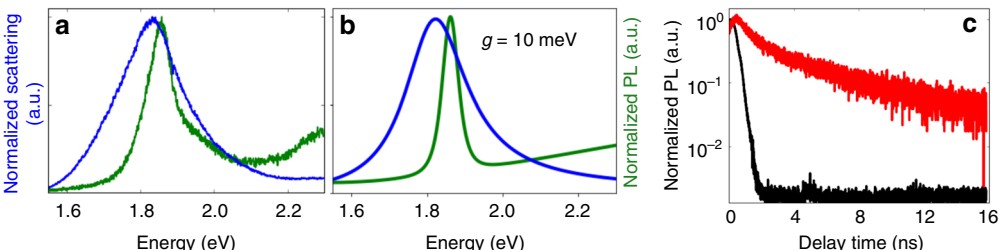

**Fig. 3** Quantum-dot / gap-plasmon systems in the weak-coupling regime. **a** Measured scattering spectrum (blue) and photoluminescence spectrum (green). **b** Theoretical scattering and photoluminescence spectra. **c** Time-resolved photoluminescence from the coupled system (black), obtained by time-correlated single-photon counting. The red curve shows a reference trace obtained from isolated quantum dots

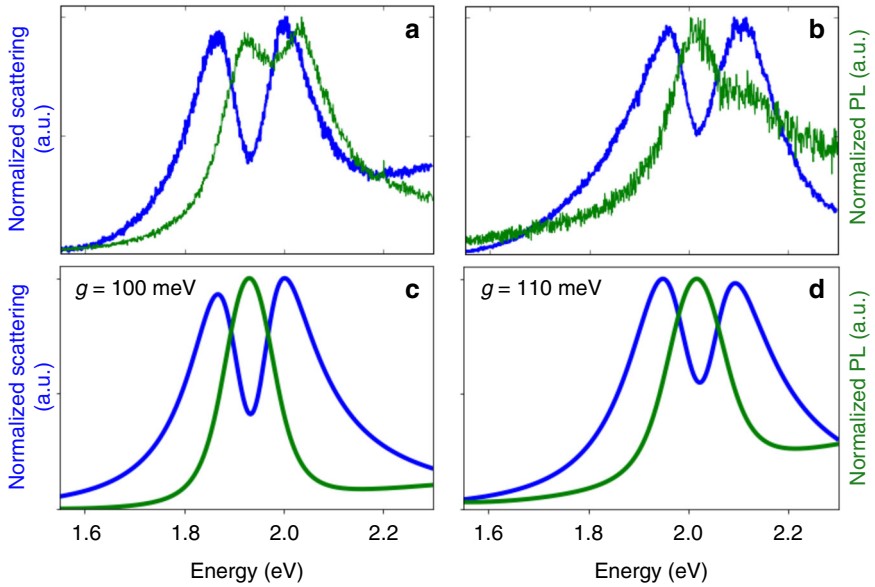

**Fig. 4** Quantum-dot / gap-plasmon systems in the intermediate-coupling regime. **a**, **b** Measured scattering spectra (blue) and photoluminescence spectra (green) before (**a**) and after (**b**) intense laser illumination. **c**, **d** Corresponding theoretical scattering and photoluminescence spectra

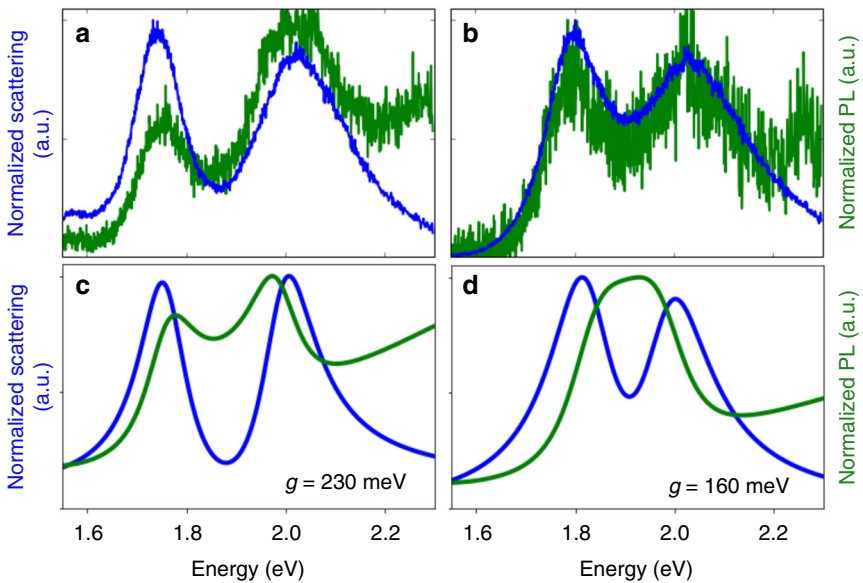

**Fig. 5** Quantum-dot / gap-plasmon systems in the strong-coupling regime. **a**, **b** Measured scattering spectra (blue) and photoluminescence spectra (green) for particles directly on a silver film (**a**) and with a 5 nm silica spacer between the particles and the silver film (**b**). **c**, **d** Corresponding theoretical scattering and photoluminescence spectra

spectrum of the coupled system still shows a Fano dip with a minimum that coincides with the maximum of the PL spectrum (See Fig. 4b, d). Since the shift in QD luminescence is due to a photochemical change, different QDs would be expected to undergo different, random shifts in their transition energies. Our observation that the PL spectrum simply shifts, without any spectral broadening, clearly indicates not only that the Fano interference arises from the QD transition, but also that it is due to coupling between the plasmon resonance and a single QD.

**Strong-coupling regime.** Finally, a small fraction (approximately 1%) of the measured samples show splitting in both the scattering

and PL spectra, providing a definitive demonstration of Rabi splitting[23]. Fig. 5a shows a representative example; we again compare the measured scattering spectrum to a fit, and the measured PL spectrum to a prediction using the parameters from the fit. The fit gives $g = 230$ meV, most likely corresponding to a QD located at the apex of a faceted AuNP. This local structure is critical to minimizing mode volume and achieving strong coupling.

An indication of the importance of the local structure is obtained by performing measurements on a sample with a 5 nm silica spacer layer between the Ag film and the nanoparticles. As shown in Fig. 5b, d, it is still possible to obtain strong coupling in

this case. This shows that the separation between the metal nanoparticle and the metal film is not a decisive factor in obtaining the field localization required for strong coupling.

In both cases, the observed splitting in the PL spectrum is larger than the splitting in the predicted spectrum. This may be due to the simplicity of the model used to predict the PL spectrum. In particular, the modeled PL spectrum corresponds to energy that is coupled from the emitter into the plasmon and subsequently radiated from the plasmon into free space; the measured spectra may also include contributions from light that is directly radiated from the emitter into free space.

## Discussion

The sensitivity to local structure explains why there is a large variation in coupling strength among the different structures in a single sample, both in our case and in previous measurements[14,24]. It also indicates that precise control over nanoscale structure will be required for high-yield fabrication of coupled plasmon–emitter systems that exhibit Fano interference or Rabi splitting. If this fabrication control can be achieved, then the systems can be developed into active devices; for example, a low-power control field can saturate the QD transition, turning off Fano interference or Rabi splitting[16,17,38], and enabling nanoscale optical modulation and quantum gates. Our results thus represent an important step towards single-photon non-linearities and quantum-optical interfaces[39] at the nanometer scale and at room temperature.

## Methods

**Simulations and data analysis**. Finite-element calculations are performed using the wave-optics module in COMSOL Multiphysics. For the quantum dot, the dielectric constant is taken to be a Lorentzian representing its ground-state transition:[26]

$$\varepsilon_{QD} = \varepsilon_{\infty} - f \frac{\omega_0^2}{\omega^2 - \omega_{QD}^2 + i\gamma_{QD}\omega} \quad (1)$$

where the background dielectric constant $\varepsilon_{\infty} = 5$, the QD resonance frequency $\omega_{QD} = 1.92$ eV, the QD decay rate $\gamma_{QD} = 0.04$–0.1 eV, and the oscillator strength $f = 0.8$, all within the range of experimentally measured values. We ignore the fine structure of the absorption spectrum, as well as absorption into higher-lying states in the QD. We also ignore effects due to the spatial dependence of the exciton wavefunction within the QD.

The gold dielectric function is based on a fit to experimental values from Olmon et al.[40], and the silver dielectric function is derived from Rakic et al.[41]. The incident electric field is polarized perpendicular to the Ag film. The simulation is done in two parts. First, the field is calculated in the frequency domain with the nanostructure absent. Second, the field is calculated with the nanostructure present, and the scattered field is obtained by subtracting the result of the first calculation from the result of the second. Finally, the scattering spectrum, $S(\omega)$, is calculated by integrating the Poynting vector of the scattered field over a surface enclosing the nanostructure.

The analytical formula for the scattering spectrum is based on treating the plasmon and exciton as two coupled oscillators:[26]

$$S(\omega) \propto \omega^4 \left| \frac{\left(\omega_{QD}^2 - \omega^2 - i\gamma_{QD}\omega\right)}{\left(\omega^2 - \omega_{SP}^2 + i\gamma_{SP}\omega\right)\left(\omega^2 - \omega_{QD}^2 + i\gamma_{QD}\omega\right) - \omega^2 g^2} \right|^2 \quad (2)$$

where $\omega_{SP}$ is the plasmon resonance frequency, $\gamma_{SP}$ is the plasmon decay rate, and $g$ is the coupling constant. In this formula and elsewhere[26], $g$ is equal to the on-resonant vacuum Rabi frequency, which differs by a factor of 2 from the definition of $g$ used in some cases in the literature[42]. When comparing to simulations, $\omega_{QD}$ and $\gamma_{QD}$ are taken to be the same as the input parameters into the simulations, $\omega_{SP}$ and $\gamma_{SP}$ are determined from a separate simulation of the metal nanostructure in the absence of the QD, and $g$ is a free parameter. This model again effectively treats the QD as a two-level system, ignoring the details of its energy-level structure and spatial variations in its coupling to electric fields. It also ignores any frequency dependence of the dielectric function of the metal across the plasmon resonance, which is an adequate approximation for photon energies far from the energies of interband transitions in the metal.

The parameters obtained from the fits to scattering spectra are used to predict PL spectra, based on a model of cavity emission for a coupled cavity–emitter

system:[42]

$$PL(\omega) = \frac{\gamma_{SP}}{\pi} \left| \frac{-i(g/2)}{\left(\frac{\gamma_{SP}+\gamma_{QD}}{4} + \frac{i(\omega_{QD}-\omega_{SP})}{2} - i(\omega - \omega_{SP})\right)^2 + g'^2} \right|^2 \quad (3)$$

where $g' = \sqrt{(g/2)^2 + (\omega_{QD}/2 - \omega_{SP}/2)^2 - (\gamma_{SP}/4 - \gamma_{QD}/4)^2}$. For the coupled plasmon–QD system, this corresponds to energy that is coupled from the QD into the plasmon resonance and is subsequently radiated by the plasmon into free space.

For comparison of predicted PL spectra to experiment, we add a quadratic background due to PL from the metal nanostructure. The parameters of this quadratic function are obtained by fitting measurements of emission from a control sample of AuNPs above an Ag film, without QDs (See Supplementary Figure 3). The amplitude of the background emission is used as an adjustable parameter to best reproduce experimental spectra.

**CdSe/CdS core/shell quantum-dot synthesis**. CdSe QDs are synthesized using a modified version of the procedure by Zhong et al.[43]. A 2.1 M selenium precursor solution is prepared by dissolving 0.1658 g of Se powder in 1 mL of trioctylphosphine (TOP). A 0.3 M cadmium precursor solution is prepared by dissolving 0.3852 g of CdO in 5 mL of oleic acid and 5 mL of octadecene, heating to 250 °C, and then maintaining the temperature at 60 °C. 0.15 mL of the Se precursor is added to 5 mL of oleylamine, heated to 90 °C, degassed for 20 min, then heated to 300 °C under nitrogen gas. 1 mL of the Cd precursor is injected, and the reaction is monitored until the desired optical properties are achieved (around 5 min). The CdSe QDs are approximately 6 nm in diameter. QDs are purified by extracting 3 times using a hexane–methanol mixture.

CdS shells are added by adapting the protocol used by Xie et al.[44]. All solutions are prepared and kept under $N_2$ for the duration of the reaction. The CdSe QDs are added to 1.5 mL of octadecene and 0.5 g of hexadecylamine and heated at 100 °C for 30 min to remove hexane. A sulfur precursor is prepared by dissolving 0.032 g of sulfur powder in 10 mL of octadecene, which is then heated to 180 °C and then held at room temperature. A cadmium precursor is prepared by dissolving 0.3204 g of CdO in 7 mL of oleic acid and 18 mL of octadecene, heating to 240 °C, then cooling to 80 °C. Shells are formed by alternating between the addition of Cd and S precursor solutions to the QDs at 235 °C at 10 min intervals, until 9 shells in total are formed. The total diameter of the QDs with the shells is approximately 11.5 nm.

**Linking of gold nanoparticles and quantum dots**. AuNPs and QDs are linked through ethyldimethylaminocarbodiimide (EDC) coupling. The synthesized CdSe/CdS core/shell QDs are functionalized with cysteamine following the procedure described by Zheng et al.[45]. CdSe/CdS QDs are purified through methanol/hexane extraction, hexane is evaporated off, and the QDs are redispersed in THF. To 0.2 g of cysteamine melted at 80 °C, 2 mL of QDs in THF solution is added and stirred for 2 h. THF is then allowed to boil off, the QDs are redispersed in water, and the QDs are then purified through dialysis against pure water.

100 nm gold nanoparticles stabilized with citrate, purchased from Sigma Aldrich, are functionalized with thioctic acid through a ligand exchange. 500 μL of 0.3 mM aqueous thioctic acid solution is added to 5 mL of as-purchased AuNPs. The pH is then raised to 8 using sodium bicarbonate. Excess thioctic acid is removed through centrifugation and redispersion in pure water.

To this solution of thioctic acid-coated AuNPs, 1 mg of EDC and 1 mg of N-hydroxysulfosuccinimide (sNHS) are added. After 20 min of stirring, QDs are added, such that the QD:AuNP molar ratio is 50:1, and the pH of the solution is raised to 8 using saturated sodium bicarbonate. The solution is then stirred overnight to allow for the covalent attachment of QDs to AuNP through amide bond formation. The resulting linked particles are centrifuged at 10,000g for 10 min and then redispersed in water, to remove the unreacted QDs.

**Ag film and sample preparation**. Smooth silver films are produced by template stripping[46]. A 100 nm silver film is deposited on a cleaned silicon wafer by thermal evaporation. Epoxy is then applied to the silver film and a clean glass slide is placed on top. The epoxy is cured in an oven at 75 °C for 30 min and is then allowed to cool to room temperature for 30 min. The glass slide and silicon wafer are pulled apart, leaving the silver film on the glass slides with an ultra-smooth exposed surface. For samples with a silica spacer layer, 5 nm of silica is deposited on the Ag film by sputtering at a deposition rate of 2 nm/min. The AuNP–QD solution is sonicated for 10 min, then a droplet of solution is placed on the Ag film. The solution is allowed to incubate for 30–40 min and is then removed from the Ag surface by blowing with air.

**Transmission electron microscopy and scanning electron microscopy**. TEM images are acquired using an FEI Morgagni M268 instrument with a Gatan Orius CCD camera, at an electron beam energy of 100 keV. STEM images are acquired using an FEI Nova NanoSEM 450 instrument, at an electron beam energy of 30 keV.

**Optical microscopy**. Scattering measurements are performed using a reflection-mode dark-field microscope objective (Nikon NA 0.9 100×). Illumination is provided by a halogen lamp. Spectra are measured using a grating spectrometer (Acton SpectraPro 500i) equipped with a CCD camera (Princeton Instruments Pixis 400). The acquisition time is set to be 5 s. Dark counts are subtracted from the measured spectra, and the spectra are then normalized by a reference spectrum measured in the absence of the gold nanoparticles and quantum dots.

For PL measurements, light from a pulsed diode laser (PicoQuant PDL-800-D) with a wavelength of 515 nm and pulse lengths of approximately 100 ps is focused through the objective to a spot size of approximately 5 μm. For most measurements, the laser repetition rate is 40 MHz and the average power entering the objective is 0.38 mW. PL spectra are measured using the same spectrometer as for scattering measurements, and time-resolved PL is measured by time-correlated single-photon counting (TCSPC), using a PD-050-CTD single-photon detector from MPD and PicoQuant PicoHarp 300 timing electronics. In order to induce irreversible photochemical changes, the nanostructures are illuminated with continuous-wave laser illumination at an average power of 20 mW for 20 s.

## Data availability

The datasets generated and analyzed during the current study are available from the corresponding author on reasonable request.

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

## Acknowledgements

The authors thank Daniel Kazal for help with thermal evaporation of Ag films, Erik Crowe for help with sputtering of silicon dioxide films, Laszlo Takacs for help with STEM imaging at the UMBC NanoImaging Facility (NIF), and Tagide DeCarvalho for help with TEM imaging at the UMBC Keith Porter Imaging Facility. This work was supported by the National Institute of Standards and Technology under Award number 14D295 and by the National Science Foundation under Award number CHE 1507462.

## Author contributions

H.L. performed deposition, optical measurements, data analysis, and simulations, and was primarily responsible for preparing the manuscript. B.S. synthesized and linked nanoparticles and performed electron-microsocope imaging. M.-C.D. supervised nanoparticle synthesis and assembly. M.P. conceived the experiment and supervised deposition, optical measurements, data analysis, and simulations. All authors contributed to experimental design, data interpretation, and preparation of the manuscript.

## Additional information

**Competing interests:** The authors declare no competing interests.

