## [Peer Review File · Nature Communications]

Reviewers' comments:

Reviewer #1 (Remarks to the Author):

This is an interesting report in a topical area. The key claim made by the authors is that they are able to observe weak, intermediate and strong coupling from single quantum dots combined with the plasmon modes associated with particle-on-film plasmonic resonances. If true such a claim makes a useful advance to the field. In general the paper is well written and the data provided are clear.

Whilst I read with interest the authors' justification that they only observe phenomena associated with single QDs, how can they be so sure? Could it be that there are two QDs adjacent to each other? Perhaps their calculations of strong coupling under such conditions would help them rule out this possibility? In my view this question needs addressing.

On a more minor note, I suggest two things.

1 - in the last sentence of the discussion, I think "the first step" should be changed to "a first step".

2 - regarding the discussion of whether the observation of two peaks is sufficient to claim strong coupling, the authors could refer to ACS Photonics, 1, p454 (2014)

Reviewer #2 (Remarks to the Author):

The manuscript "Strong coupling between single quantum dots and gap plasmons at room temperature" by Leng et al; reports measurements of scattering and photoluminescence (PL) from individual coupled plasmon emitter systems [a single colloidal quantum dot (QD)] in the gap between a gold nanoparticle and a silver film.

The measurements demonstrate experimentally effects predicted theoretically in some theoretical articles (see comments below).

In particular, they observe weak coupling (the Purcell effect), intermediate coupling (Fano interference), and strong coupling (Rabi splitting) at room temperature.

It is of great interest to provide both scattering and PL experimental spectra because, as remarked in the article, the two spectra can give different information allowing for a better characterization of the various coupling regimes.

For example, as pointed out by the authors, the simultaneous measurement of scattering and PL spectra allows Fano interference to be distinguished from geometric properties of the metal nanostructure that could give rise to a double-peak scattering spectrum.

In summary, in my opinion, the work appears interesting, original, and can be worth of publication in this journal.

Nonetheless, I have some comments that I think have to be addressed:

1) First of all I think that the manuscript and, as consequence, the bibliography have to be enriched by more key articles about the field.

In particular,

- line 25 citation 2.

In addition to the reference 2, I suggest to cite at least the paper
"Nanopolaritons: Vacuum Rabi Splitting with a Single Quantum Dot in the Center of a Dimer Nanoantenna"

By: Savasta, Salvatore et al., ACS Nano, 2010, 4 (11), pp 6369–6376 DOI: 10.1021/nn100585h
where the authors theoretically demonstrate, with scattering calculations that a system constituted by a single quantum dot placed in the gap between two metallic nanoparticles can display the vacuum Rabi splitting.

- line 54 ...

Regarding the Fano interference effect I suggest to add some references. One of them is surely
"Quantum Plasmonics with Quantum Dot-Metal Nanoparticle Molecules: Influence of the Fano Effect on Photon Statistics"

Ridolfo, A. et al. PHYSICAL REVIEW LETTERS 105, 26, 263601 (2010)

where the authors study theoretically the quantum optical properties of hybrid molecules composed

of an individual quantum dot and a metallic nanoparticle showing that the coupling between the two systems gives rise

to a Fano interference effect which strongly influences the quantum statistical properties of the scattered photons.

- line 187

The saturation of the QD transition, turning off Fano interference in order to enable nanoscale optical modulation and quantum gates was first theoretically discussed in

Ridolfo, A. et al. PHYSICAL REVIEW LETTERS 105, 26, 263601 (2010).

In summary, I suggest a more accurate introduction for better describe to the state-of-art of research in the field and in order to better frame the results obtained by authors.

In my opinion the two references cited above are mandatory.

In addition,

I may suggest some other interesting references that can be added leaving the authors free to decide if or what of them have been cited. The list is obviously not exhaustive and is also limited to my knowledge of literature.

-Andersen et al, Strongly modified plasmon–matter interaction with mesoscopic quantum emitters Nature Physics volume 7, pages 215–218 (2011);

-Tame et al, Quantum plasmonics Nature Physics volume 9, pages 329–340 (2013);

-Cacciola et al, Ultrastrong coupling of Plasmons and Excitons in a Nanoshell ACS NANO 8, 11, 11483-11492 (2014);

-Jun et al, Strong Modification of Quantum Dot Spontaneous Emission via Gap Plasmon Coupling in Metal Nanoslits J. Phys. Chem. C, 2010, 114 (16), pp 7269–7273

DOI: 10.1021/jp9083376 2009;

-Curto et al, Unidirectional Emission of a Quantum Dot Coupled to a Nanoantenna, Science Vol. 329, Issue 5994, pp. 930-933 (2010).

2) Most of the Figures must be improved:

- Fig 3 (c) the labels in y axis have to be modified using the exponential notation with 10 instead of

e ($1e-1$ and $1e-2$ must become 10^{-1} and 10^{-2} respectively).

- Fig 3 the measure units are better put between round braces [e.g.: Energy, eV becomes Energy (eV)]

- Fig 4 see comments on Figure 3.

- I suggest a rearrangement of all figures according to the comments already done for Fig. 3 and/or to the style of the journal.

4)

In my opinion, the authors have to comment more accurately on the discrepancies between the experimental and theoretical PL spectra in Fig 4 (a, c) and Fig 5(a,c).

They theoretically calculated the PL spectra using a model of cavity emission for a coupled cavity-emitter system Ref [27] (line 217).

In my opinion, the discrepancies between the experimental and theoretical spectra are also due to the fact that the model doesn't take into account the incoherent processes of photoluminescence and the effects of thermalization focussing only on spontaneous emission.

5) (optional correction)

line 218

formula (3) rhs:

For clarity and in order to be little more concise, in my opinion, it is better to cancel $-i$ from the numerator in the absolute value.

Reviewer #3 (Remarks to the Author):

Report on manuscript entitled "Strong coupling between single quantum dots and gap plasmons at room temperature" by Leng and coworkers.

The authors address the control of spontaneous emission of single quantum emitters at room temperature using plasmonic structures that provide ultra-low optical volume. This is a timely and important topic with only very few works claiming strong coupling so far, sometimes in an unconvincing manner. I have been following the field carefully and I am happy to finally read a manuscript that provides definitive evidence for strong coupling regime. With previous claims for strong coupling, a critical question came from the use of elastic scattering as a probe. Indeed, the light elastically scattered from a two-level system coupled to a cavity mode, whether in reflectivity or in transmission, results from the interferences between the incident light field and the light scattered by the system. These interferences can lead to the appearance of double peaks in scattered spectra while the system is in the weak coupling regime. This is observed in high cooperativity single quantum dot cavity devices operating in the bad cavity limit (see for instance Nature Nanotechnology 12, 663 (2017)) but is also well known in the field of cavity polaritons where 2D quantum well excitons are coupled to planar cavities. See for instance Solid State Communications 93 (9), 733-739 (1995) where the different values of Rabi splitting when measuring reflectivity, transmission or absorption are discussed.

To provide definitive evidence for strong coupling regime, one should not only observe a doublet in scattered light but also in photoluminescence spectroscopy, when using a non-resonant excitation.

The only previous work on this topic is reference 16 where 4 peaks were observed, almost independently of the quantum dot-plasmon coupling, rendering any claim very questionable.

The present work reports the first compelling demonstration of strong coupling regime in photoluminescence spectroscopy. The authors present measurements using both elastic scattering and photoluminescence on various quantum dot-antenna devices and show the critical differences between both techniques. The paper is very well written, with strong claims supported both by measurements and calculations. It represents an important contribution to this timely topic and I think it should be published in Nature Communications. I actually think that this work would be perfectly suited for a higher profile journal such as Nature Photonics.

I list some comments below that the authors should address before publication.

- In the introduction, line 26, the references to self-assembled quantum dot papers for weak and strong coupling refers only to the US work. However, the first demonstrations (1 US, 1 French, 1 Germany) of strong coupling regime with three different dielectric cavity systems were almost simultaneous. Please correct.

- Similarly, line 29-30, the authors give only US references for the control of reflectivity by a single self-assembled quantum dot and the application to the development of optical gates. Please, be fair and include other salient works showing elastic scattering with clear double peaks and their use for photon filters, photon-photon gates etc. The authors must be aware of other important contributions in this field outside the US.

- Line 47: the use of a coupled oscillator model to account for the observation could be confusing considering that text book model for the system under study is the Jaynes Cumming model, coupling a two-level system (a spin) to a harmonic oscillator. Please indicate that both models give the same predictions at low excitation when there is no saturation effect from the two-level system.

- Line 96-99: "small fraction", "even smaller fraction". Since the authors have made systematic measurements on many structures, they should give quantitative values: is it 1% of the structures that show intermediate or strong coupling regime? Do all structures with a quantum dot at the apex of a facet show intermediate or strong coupling?

- Line 100: strong localization is used to explain that the observed splitting can only be due to a single quantum dot. This is a strong statement that is not really supported. Could the author justify? This could be done either by discussing the respective plasmon mode volume and quantum dot nanocrystal size or by providing a $g(2)$ measurement.

- Line 112: "in the majority of the cases": please be quantitative. Give percentage.

- Line 139-140: the attribution of the shoulder in the spectrum to charged exciton is not supported by any observation here. The authors only provide an old reference. Could they indicate what in their system show that this is indeed the case (and not a biexciton for instance?)

- Lines 160-161: this the only weak point of the paper: the absence of direct experimental evidence that the coupling is obtained by a single emitter only. Here the authors indicate that power dependent spectral shift of different quantum dots are very likely to be different. I don't really see

why, but if this is actually the case, then the authors should provide experimental data supporting their claims: show the spectral modification induced by power measured on various single quantum dots when not coupled to the plasmonic structure.

- In general, the paper would strongly benefit from a $g(2)$ measurement. Why is this measurement not provided? The authors perform time resolved measurements with single photon detectors, why not $g(2)$?

- Line 167-170: the authors “repeat” the same measurement with an additional layer of silica. I guess these are different quantum dots and plasmonic antennas. Since the coupling varies a lot even in the absence of silica from dot to dot, going from the weak coupling to strong coupling, I don’t think that the data provided with silica strengthen the demonstration.

Indeed, consider Figures 5b and 5d: the agreement between the theoretical (5d) and measured (5b) data with silica, is not good. Experimentally, the doublet is as strong in photoluminescence than in resonant scattering. This indicates a much stronger coupling strength than the one used theoretically in figure 5d. How did the author choose g to account for figure 5b?

With the randomness of the technology (variation of the mode volume and quantum dot position), comparing one random measurement with a silica layer to another without silica does not strengthen the demonstration. A valid demonstration would require comparing many devices with and without silica, and observe that on average, the largest coupling strength g is higher without silica. Can the authors provide such data?

We thank the reviewers for their thorough and insightful comments regarding the manuscript. We are glad that all reviewers consider the results to be novel and significant and suitable for publication in *Nature Communications*. We have revised the manuscript in response to each of the comments, and we are grateful that the revisions have strengthened the manuscript. A detailed response to the comments is below.

Reviewer #1:

This is an interesting report in a topical area. The key claim made by the authors is that they are able to observe weak, intermediate and strong coupling from single quantum dots combined with the plasmon modes associated with particle-on-film plasmonic resonances. If true such a claim makes a useful advance to the field. In general the paper is well written and the data provided are clear.

Whilst I read with interest the authors' justification that they only observe phenomena associated with single QDs, how can they be so sure? Could it be that there are two QDs adjacent to each other? Perhaps their calculations of strong coupling under such conditions would help them rule out this possibility? In my view this question needs addressing.

This is an important point, also raised by reviewer #3. We have revised the manuscript to add additional evidence and clarify how our experiments and simulations show that only a single QD is involved in coupling to the plasmon resonance:

(a) Finite-element calculations show that the field within the gap between a faceted gold nanoparticle (as required for intermediate or strong coupling) and a silver film extends over a lateral area that can accommodate only a single QD. We have added a discussion of this point to the manuscript and a figure to the Supplementary Information.

(b) The PL linewidths and the QD linewidths used in fitting the scattering spectra are consistent with the inhomogeneous linewidth of a single QD, and do not show any of the inhomogeneous broadening that would be expected if multiple QDs were involved. We have clarified this point in the text.

(c) In Figure 4, we show the change in scattering and PL spectra that arise when the QD undergoes irreversible photo-oxidation. The QD undergoes a spectral shift; since this shift is due to random photochemical events, two or more QDs are not expected to undergo identical shifts. However, the measurement shows a simple shift of the PL spectrum, without any spectral broadening, thereby indicating that a single QD is involved. We have clarified this point in the text, as well.

On a more minor note, I suggest two things.

1 - in the last sentence of the discussion, I think "the first step" should be changed to "a first step".

We have changed the sentence to read “Our results thus represent an important step towards single-photon nonlinearities and quantum-optical interfaces at the nanometer scale and at room temperature.”

2 - regarding the discussion of whether the observation of two peaks is sufficient to claim strong coupling, the authors could refer to ACS Photonics, 1, p454 (2014)

This reference has been added to the manuscript.

Reviewer #2

The manuscript "Strong coupling between single quantum dots and gap plasmons at room temperature" by Leng et al; reports measurements of scattering and photoluminescence (PL) from individual coupled plasmon emitter systems [a single colloidal quantum dot (QD)] in the gap between a gold nanoparticle and a silver film. The measurements demonstrate experimentally effects predicted theoretically in some theoretical articles (see comments below). In particular, they observe weak coupling (the Purcell effect), intermediate coupling (Fano interference), and strong coupling (Rabi splitting) at room temperature. It is of great interest to provide both scattering and PL experimental spectra because, as remarked in the article, the two spectra can give different information allowing for a better characterization of the various coupling regimes. For example, as pointed out by the authors, the simultaneous measurement of scattering and PL spectra allows Fano interference to be distinguished from geometric properties of the metal nanostructure that could give rise to a double-peak scattering spectrum.

In summary, in my opinion, the work appears interesting, original, and can be worth of publication in this journal.

We thank the reviewer for the positive assessment of the manuscript.

Nonetheless, I have some comments that I think have to be addressed:

1) First of all I think that the manuscript and, as consequence, the bibliography have to be enriched by more key articles about the field.

In particular,

- line 25 citation 2.

In addition to the reference 2, I suggest to cite at least the paper "Nanopolaritons: Vacuum Rabi Splitting with a Single Quantum Dot in the Center of a Dimer Nanoantenna"

By: Savasta, Salvatore et al., ACS Nano, 2010, 4 (11), pp 6369–6376 DOI: 10.1021/nn100585h where the authors theoretically demonstrate, with scattering calculations that a system constituted by a single quantum dot placed in the gap between two metallic nanoparticles can display the vacuum Rabi splitting.

- line 54 ...

Regarding the Fano interference effect I suggest to add some references. One of them is surely "Quantum Plasmonics with Quantum Dot-Metal Nanoparticle Molecules: Influence of the Fano Effect on Photon Statistics" Ridolfo, A. et al. PHYSICAL REVIEW LETTERS 105, 26, 263601 (2010) where the authors study theoretically the quantum optical properties of hybrid molecules composed of an individual quantum dot and a metallic nanoparticle showing that the coupling between the two systems gives rise to a Fano interference effect which strongly influences the quantum statistical properties of the scattered photons.

- line 187

The saturation of the QD transition, turning off Fano interference in order to enable nanoscale optical modulation and quantum gates was first theoretically discussed in Ridolfo, A. et al. PHYSICAL REVIEW LETTERS 105, 26, 263601 (2010).

In summary, I suggest a more accurate introduction for better describe to the state-of-art of research in the field and in order to better frame the results obtained by authors.

In my opinion the two references citated above are mandatory.

In addition, I may suggest some other interesting references that can be added leaving the authors free the decide if or what of them have been cited. The list is obviously not exhaustive and is also limited to my knowledge of literature.

-Andersen et al, Strongly modified plasmon–matter interaction with mesoscopic quantum emitters Nature Physics volume 7, pages 215–218 (2011);

-Tame et al, Quantum plasmonics Nature Physics volume 9, pages 329–340 (2013);

-Cacciola et al, Ultrastrong coupling of Plasmons and Excitons in a Nanoshell ACS NANO 8, 11, 11483-11492 (2014);

-Jun et al, Strong Modification of Quantum Dot Spontaneous Emission via Gap Plasmon Coupling in Metal Nanoslits J. Phys. Chem. C, 2010, 114 (16), pp 7269–7273 DOI: 10.1021/jp9083376 2009;

-Curto et al, Unidirectional Emission of a Quantum Dot Coupled to a Nanoantenna, Science Vol. 329, Issue 5994, pp. 930-933 (2010).

We have added the first two references indicated as mandatory and two of the additional suggested references. There have been many important papers published on plasmon-emitter coupling, and it would be impossible to include an exhaustive list in this article. We have therefore focused on those most directly related to the current work, and we thank the reviewer for the additional suggestions.

2) Most of the Figures must be improved:

- Fig 3 (c) the labels in y axis have to be modified using the exponetial notation with 10 instead of e (1e-1 and 1e-2 must become 10^{-1} and 10^{-2} respectively).

- Fig 3 the measure units are better put between round braces [e.g.: Energy, eV becomes Energy (eV)]

- Fig 4 see comments on Figure 3.

- I suggest a rearrangement of all figures according to the comments already done for Fig. 3 and/or to the style of the journal.

The figures have been modified as suggested.

4) In my opinion, the authors have to comment more accurately on the discrepancies between the experimental and theoretical PL spectra in Fig 4 (a, c) and Fig 5(a,c). They theoretically calculated the PL spectra using a model of cavity emission for a coupled cavity-emitter system Ref [27] (line 217). In my opinion, the discrepancies between the experimental and theoretical spectra are also due to the fact that the model doesn't take into account the incoherent processes of photoluminescence and the effects of thermalization focussing only on spontaneous emission.

We thank the reviewer for this insight. There are indeed several simplifications in the model that can account for the difference between the measured and calculated PL spectra. As well as those mentioned by the reviewer, the model does not take into account fine structure in the quantum-dot energy levels or the finite size of the exciton wavefunction in the quantum dot. In addition, the theoretical model for PL that we use is adapted from cavity QED, where it is used to describe end emission from a cavity; that is, light that is coupled from the emitter into the cavity and then leaks out the end mirrors. We believe that this is the most appropriate model for our observations, because the majority of the detected light couples from the quantum dots into the plasmon resonance, and is then radiated by the plasmon into free space. However, there may also be a contribution to the spectrum from “side emission,” or direct radiation from the quantum dot into free space.

We have added these points to the manuscript.

5) (optional correction) line 218 formula (3) rhs: For clarity and in order to be little more concise, in my opinion, it is better to cancel -i from the numerator in the absolute value.

We thank the reviewer for the opportunity to simplify our formula. We have made the corresponding change.

Reviewer #3:

The authors address the control of spontaneous emission of single quantum emitters at room temperature using plasmonic structures that provide ultra-low optical volume. This is a timely and important topic with only very few works claiming strong coupling so far, sometimes in an unconvincing manner. I have been following the field carefully and I am happy to finally read a manuscript that provides definitive evidence for strong coupling regime. With previous claims for strong coupling, a critical question came from the use of elastic scattering as a probe. Indeed, the light elastically scattered from a two-level system coupled to a cavity mode, whether in reflectivity or in transmission, results from the interferences between the incident light field and the light scattered by the system. These interferences can lead to the appearance of double peaks in scattered spectra while the system is in the weak coupling regime. This is observed in high cooperativity single quantum dot cavity devices operating in the bad cavity limit (see for instance Nature Nanotechnology 12, 663 (2017)) but is also well known in the field of cavity polaritons where 2D quantum well excitons are coupled to planar cavities. See for instance Solid State Communications 93 (9), 733-739 (1995) where the different values of Rabi splitting when measuring reflectivity, transmission or absorption are discussed.

To provide definitive evidence for strong coupling regime, one should not only observe a doublet in scattered light but also in photoluminescence spectroscopy, when using a non-resonant excitation. The only previous work on this topic is reference 16 where 4 peaks were observed, almost independently of the quantum dot-plasmon coupling, rendering any claim very questionable.

The present work reports the first compelling demonstration of strong coupling regime in photoluminescence spectroscopy. The authors present measurements using both elastic scattering and photoluminescence on various quantum dot-antenna devices and show the critical differences between both techniques. The paper is very well written, with strong claims supported both by measurements and calculations. It represents an important contribution to this timely topic and I think it should be published in Nature Communications. I actually think that this work would be perfectly suited for a higher profile journal such as Nature Photonics.

We thank the reviewer for the positive assessment of our work, and we hope that the readers of Nature Communications will find it similarly significant.

I list some comments below that the authors should address before publication.

- In the introduction, line 26, the references to self-assembled quantum dot papers for weak and strong coupling refers only to the US work. However, the first demonstrations (1 US, 1 French, 1 Germany) of strong coupling regime with three different dielectric cavity systems were almost simultaneous. Please correct

The reference in the original manuscript was to a review paper. We have added references to the three original research papers.

- Similarly, line 29-30, the authors give only US references for the control of reflectivity by a single self-assembled quantum dot and the application to the development of optical gates. Please, be fair and include other salient works showing elastic scattering with clear double peaks and their use for photon filters, photon-photon gates etc. The authors must be aware of other important contributions in this field outside the US.

We apologize for omitting these references, and have added them to the manuscript.

- Line 47: the use of a coupled oscillator model to account for the observation could be confusing considering that text book model for the system under study is the Jaynes-Cumming model, coupling a two-level system (a spin) to a harmonic oscillator. Please indicate that both models give the same predictions at low excitation when there is no saturation effect from the two-level system.

We have added to the manuscript a sentence clarifying this point.

- Line 96-99: “small fraction”, “even smaller fraction”. Since the authors have made systematic measurements on many structures, they should give quantitative values: is it 1% of the structures that show intermediate or strong coupling regime? Do all structures with a quantum dot at the apex of a facet show intermediate or strong coupling?

We are not able to directly image the location of the QD in the gap relative to facets on the nanocrystal, so we can give only the fraction of measured particles that exhibit weak, intermediate or strong coupling. We have added these approximate percentages to the text.

- Line 100: strong localization is used to explain that the observed splitting can only be due to a single quantum dot. This is a strong statement that is not really supported. Could the author justify? This could be done either by discussing the respective plasmon mode volume and quantum dot nanocrystal size or by providing a $g(2)$ measurement.

We thank the reviewer for the opportunity to strengthen this point. We have used our finite-element calculations to estimate the lateral dimensions over which the localized field extends within the gap between the gold nanoparticle and the silver film. We show that the localization is such that only one QD could be located within the high-field region. We have added text to the manuscript and a figure to the Supplementary Information discussing this point.

- Line 112: “in the majority of the cases”: please be quantitative. Give percentage.

As mentioned above, we have added estimates of the percentages based on the fraction of observed structures that exhibit intermediate or strong coupling.

- Line 139-140: the attribution of the shoulder in the spectrum to charged exciton is not supported by any observation here. The authors only provide an old reference. Could they indicate what in their system show that this is indeed the case (and not a biexciton for instance?)

The nature of the emitting state responsible for the shoulder does not affect the conclusions in our paper. We have therefore indicated in the revised manuscript that it could be due to charged-exciton or biexciton emission. The old reference is still valid, but we have added an additional reference to show that the appearance of this shoulder is expected under strong excitation (as is the case for a QD in the high-field region of a metal nanostructure).

- Lines 160-161: this the only weak point of the paper: the absence of direct experimental evidence that the coupling is obtained by a single emitter only. Here the authors indicate that power dependent spectral shift of different quantum dots are very likely to be different. I don't really see why, but if this is actually the case, then the authors should provide experimental data supporting their claims: show the spectral modification induce by power measured on various single quantum dots when not coupled to the plasmonic structure.

The change in the QD transition energy is not a power-dependent spectral shift, but rather an irreversible change due to photochemical modification of the QD. The system is exposed to intense laser illumination, which produces the changes, but the measurements are taken after the illumination, under the same intensity as the other reported measurements. We apologize that this point was not clear in the original paper, and have revised the wording to clarify.

These photochemical shifts are random, and are thus unlikely to be the same for multiple QDs. We thus believe that observing a spectrum that corresponds to a simple shift in the PL peak, without any increase in the PL linewidth, is a strong indication that a single QD is involved.

We note also that the PL linewidths throughout the manuscript are consistent with a single QD, and do not show the inhomogeneous broadening that would be expected for multiple QDs. We have added text to the manuscript to clarify this point, as well. We believe that these factors, together with the estimate of field confinement described above, provide good evidence that the coupling is due to a single QD.

- In general, the paper would strongly benefit from a $g^{(2)}$ measurement. Why is this measurement not provided? The authors perform time resolved measurements with single photon detectors, why not $g^{(2)}$?

For the system studied, a $g^{(2)}$ measurement cannot be used to distinguish single QDs from multiple QDs. Typically, $g^{(2)}$ measurements are used to identify single QDs by observing that the photon autocorrelation function approaches zero for zero time delay ($g^{(2)}(0) = 0$). This indicates that only one photon is emitted at a time, a sign that a single emitter is being observed. However, for colloidal QDs, single-photon emission occurs because of rapid Auger recombination of the biexciton state. A single band-edge exciton in the QD will undergo primarily radiative

recombination with a lifetime on the order of 10 ns; if more than one exciton is present, recombination occurs through a rapid, non-radiative Auger process with a lifetime on the order of 100 ps or less. Since the non-radiative Auger process for two excitons is much faster than the radiative recombination process, the photoluminescence quantum yield for biexcitons is nearly zero; i.e., a quantum dot typically emits a photon only when a single exciton is present.

However, when the QD is located in the high-field region of a metal nanostructure, its radiative lifetime decreases dramatically due to the Purcell effect. As shown in Figure 3c, the modified radiative lifetime for a QD coupled to our plasmonic nanostructures is comparable to the Auger lifetime. This means that biexcitons are now likely to recombine radiatively, rather than through the non-radiative Auger process. Since biexcitons can recombine nearly as efficiently as single excitons, the QD is no longer restricted to emitting a single photon at a time. The autocorrelation function thus no longer approaches zero at zero time delay. This effect has been observed in several published experiments. (See, for example, H. Naki et al., J. Pys. Chem. C 2011, 115, 23299; S. J. LeBlanc et al., Nano Lett. 2013, 13, 1662; and Y. S. Park et al., Phys. Rev. Lett. 2013, 110, 117401.) This means that $g^{(2)}$ measurements no longer provide a means of distinguishing single QDs from multiple QDs.

- *Line 167-170: the authors “repeat” the same measurement with an additional layer of silica. I guess these are different quantum dots and plasmonic antennas. Since the coupling varies a lot even in the absence of silica from dot to dot, going from the weak coupling to strong coupling, I don’t think that the data provided with silica strengthen the demonstration.*

o Indeed, consider Figures 5b and 5d: the agreement between the theoretical (5d) and measured (5b) data with silica, is not good. Experimentally, the doublet is as strong in photoluminescence than in resonant scattering. This indicates a much stronger coupling strength than the one used theoretically in figure 5d. How did the author choose g to account for figure 5b?

o With the randomness of the technology (variation of the mode volume and quantum dot position), comparing one random measurement with a silica layer to another without silica does not strengthen the demonstration. A valid demonstration would require comparing many devices with and without silica, and observe that on average, the largest coupling strength g is higher without silica. Can the authors provide such data?

The reviewer is correct that two measurements do not allow for systematic comparison of the differences that occur when a silica film is present. Figures 5b,d are intended only to show that strong coupling is possible even for larger gaps between the gold nanoparticle and the silver film. The gap size is thus not the decisive factor in obtaining strong coupling; rather, it must be the local structure of the metal nanoparticle that is important. We apologize that this point was not clear in the original manuscript, and we have revised the wording to make it more clear.

We also agree that the predicted PL spectra do not show splitting as strong as the experimental spectra. To clarify, the predicted spectra are obtained by first fitting the measured scattering spectra, and then using the parameters from that fit to predict the PL spectra. We believe that the differences between the predicted and measured PL spectra arise primarily from the fact that our model assumes that all of the observed photons come from energy that is coupled from the QD to the

metal nanostructure and subsequently radiated into free space. In practice, the measured spectra likely also include some light that is radiated directly from the QDs into free space. We have modified the manuscript to clarify this point.

REVIEWERS' COMMENTS:

Reviewer #1 (Remarks to the Author):

I have now looked at the responses of the authors to the questions raised by myself and the other reviewers. I am happy that all points have been sufficiently addressed and am happy to see the paper published in Nat Comm.

Reviewer #3 (Remarks to the Author):

The authors have correctly addressed all my questions and comments. I recommend publication in Nature Communications without further changes.